# Effectiveness of pneumococcal and influenza vaccines to prevent serious health complications in adults with chronic liver disease: a protocol for a systematic review

Suvi Härmälä,[1] Constantinos Parisinos,[1] Laura Shallcross,[1] Alastair O'Brien,[2] Andrew Hayward[3]

[1]Institute of Health Informatics, University College London, London, UK
[2]Division of Medicine, University College London, London, UK
[3]Institute of Epidemiology and Health Care, University College London, London, UK

**Correspondence to**
Suvi Härmälä;
suvi.harmala.14@ucl.ac.uk

## ABSTRACT

**Introduction** In advanced chronic liver disease, diseases caused by common bacteria *Streptococcus pneumoniae* or influenza virus put people at an increased risk of serious health complications and death. The effectiveness of the available vaccines in reducing the risk of poor health outcomes, however, is less clear.

**Methods and analysis** We will search Medline (Ovid), Embase (Ovid), PubMed and Cochrane Central Register of Controlled Trials for published reports on randomised controlled trials and observational studies on the effectiveness of pneumococcal and influenza vaccines in people with chronic liver disease. Two independent reviewers will screen the studies for eligibility, extract data and assess study quality and risk of bias. Random effects meta-analyses will be performed as appropriate.

**Ethics and dissemination** Formal ethical approval is not required, as no primary data will be collected for this study. We will publish results of this study in relevant peer-reviewed medical journal or journals. Where possible, the study results will also be presented as posters or talks at relevant medical conferences and meetings.

**PROSPERO registration number** CRD42017067277.

### Strengths and limitations of this study

► This study protocol follows the recommendations by the Preferred Reporting Items for Systematic Review and Meta-Analysis Protocols.
► This study protocol has been prospectively registered in the International Prospective Register of Systematic Reviews.
► Our comprehensive search strategy will minimise the risk of missing relevant studies, in particular, those with a randomised design.
► The selection of studies, data extraction, the risk of bias and quality of evidence assessments will be conducted by two independent authors.
► Inclusion of studies with a non-randomised design may decrease the overall quality of the body of evidence for the study outcomes.

## BACKGROUND
### Burden of pneumococcal and influenza infections in chronic liver disease

In advanced chronic liver disease, as the immune function progressively deteriorates, diseases caused by common bacteria *Streptococcus pneumoniae* or influenza virus can lead to serious health complications and death.

In Spain in 2011, the population-level annual incidence rate for pneumococcal pneumonia-related hospitalisation in adults with liver disease was estimated at approximately 540 per 100 000 compared with approximately 6 per 100 000 without at-risk conditions.[1] Adults with liver disease were over 50 times more likely to be hospitalised for pneumococcal pneumonia than adults

without at-risk conditions.[1] Similarly, in England in 2008/2009, for invasive pneumococcal disease (IPD), such as meningitis, bacteraemia and sepsis, the annual incidence of hospitalisation in adults with liver disease was estimated at about 100 per 100 000 compared with about 8 per 100 000 in the healthy population.[2] Approximately 37% of liver disease patients hospitalised for IPD died compared with 5% of patients without underlying risk conditions.[2] Chronic liver disease patients were over 30 times more likely to be admitted to hospital and 10 times more likely to die during the IPD-related hospitalisation than adults without at-risk conditions.[2]

Although there are no population-level estimates of severe influenza incidence in people with chronic liver disease, evidence suggests that liver disease patients are at an increased risk of health complications from influenza. A twofold increased risk of influenza admission was observed in liver disease patients at

19 hospitals in Russia, Turkey, China and Spain during the 2013/2014 season.[3] Similarly, an analysis of data on laboratory-confirmed influenza cases collected from several WHO member states during the 2009 influenza A (H1N1) pandemic found liver disease patients to have a greater than fivefold increased risk of influenza-related hospitalisation and over 17-fold increased risk of death compared with that of healthy individuals.[4] Furthermore, influenza infection, while not directly targeting the liver, may cause collateral transient liver damage[5] and trigger hepatic decompensation (such as ascites and hepatic encephalopathy) in liver disease patients.[6]

### Types of pneumococcal and influenza vaccines, vaccination policy and vaccine uptake

Two types of vaccines, polysaccharide pneumococcal vaccines including serotypes of *S. pneumoniae* (PPV23) and conjugate pneumococcal vaccines including 7 (PCV7), 10 (PCV10) or 13 (PCV13) *S. pneumoniae* serotypes, are available to protect against pneumococcal infection. None of these contains live bacteria. The most commonly used influenza vaccines are injectable, inactivated vaccines that contain either inactivated whole influenza virus or split or subunit virus products. These vaccines protect either against two influenza A (H1N1 and H3N2) strains and one influenza B strain (trivalent vaccines) or two influenza A and two influenza B strains (quadrivalent vaccines). New vaccines are developed every year to protect against the prevailing strains of the upcoming influenza season and a yearly vaccination is recommended to ensure continued protection. Live attenuated influenza vaccines exist, however, these may not be suitable for people with chronic comorbidities.[7] Immune response defects associated with advanced liver disease[8–11] may also dampen the response to vaccines.

The majority of European countries recommend both adult influenza (29/29 countries) and pneumococcal vaccination (22/29 countries) for specific high-risk groups,.[12] Chronic liver disease patients, however, may not be included in these high-risk target groups in all countries. While 90% (27/30) of the countries recommended influenza immunisation for people with liver disease during the 2014–2015 influenza season,[13] only 43% (6/14) of countries surveyed in 2005 recommended pneumococcal vaccination for patients with chronic liver disease.[14] Moreover, while the adult vaccination recommendations for influenza are supported by official funding mechanisms in most countries (21/29), the cost of the pneumococcal vaccination is covered by only half (11/22).[12]

Uptake of influenza vaccine in people with chronic diseases in general is poor. The median coverage rate across Europe in the 2014/2015 season was less than 50% (only 7/30 countries were able to provide separate coverage data for individuals with chronic medical conditions).[13] The situation in patients with liver disease as a separate group seems no different with less than 50% of working-age patients with liver disease in the UK covered

by the influenza vaccine in 2015/2016 season.[15] Although fewer data exist concerning the uptake of pneumococcal adult vaccination,[12] its uptake in patients with liver disease is not likely to be any higher than that of influenza vaccine.

### Rationale for the review

Approximately 29 million people in Europe alone are affected by some form of liver disease.[16] Worryingly, due to the increasing prevalence of obesity and persistence of other liver damage risk factors such as alcohol abuse and hepatitis infections, this number is expected to grow.[16] While evidence suggests that following infection with influenza or *S. pneumoniae*, people with liver disease have a higher than average risk of poor health outcomes, the effectiveness of the vaccines in reducing this risk is less clear and warrants further investigation.

To our knowledge, no systematic review to date has investigated the effects of pneumococcal and influenza vaccines in preventing poor health outcomes in chronic liver disease. The present review intends to fill this gap by providing a systematic synthesis of the available evidence. The results of this review may inform future vaccination strategies and help improve vaccination coverage. This, in turn, may have a positive impact on both the number of influenza and pneumococcal disease-related hospital admissions and patient outcomes in liver disease.

### OBJECTIVES

The aim of this review is to assess the effectiveness of pneumococcal and influenza vaccines to prevent serious health complications in adults with chronic liver disease.

The objectives are:
► To assess the effectiveness of pneumococcal and influenza vaccines to prevent hospitalisation in adults with chronic liver disease;
► To assess the effectiveness of pneumococcal and influenza vaccines to prevent death in adults with chronic liver disease;
► To assess the effects of pneumococcal and influenza vaccines for eliciting a serological response in adults with chronic liver disease.

### METHODS

This study protocol follows the recommendations by the Preferred Reporting Items for Systematic Review and Meta-Analysis Protocols (PRISMA-P) 2015.[17]

### Eligibility criteria
#### Types of studies

We will include all randomised clinical trials, cohort (with comparison group/s) and case–control studies that investigate the effectiveness of pneumococcal or influenza vaccines for preventing hospitalisation or death in adults with chronic liver disease. We will also

include all randomised clinical trials, cohort (with or without comparison group/s) and case–control studies that report the serological response to one or both of these vaccines in adults with chronic liver disease. We will only include published studies in English language and studies that have been published or accepted for publication. We will exclude review articles, case reports, cross-sectional studies, animal studies, editorials, clinical guidelines and any studies that have been fully or partially retracted from publication. Patients included in multiple studies will be reported only once.

## Types of participants

We will include studies that enrol adult patients who are at least 18 years old and have chronic liver disease of any severity (non-cirrhotic, cirrhotic) or aetiology (viral, alcoholic, non-alcoholic fatty liver, autoimmune).

## Types of interventions

We will include studies that investigate the effects of a conjugate or polysaccharide pneumococcal vaccine (against *S. pneumoniae*) and/or an inactivated (whole virus, split virus or subunit), injectable influenza vaccine. Vaccines can be adjuvanted, intradermal and of any dose. We will exclude live, recombinant, virosomal and experimental vaccines.

## Types of comparators

We will include studies comparing one or both of the vaccines of interest to a placebo, an alternative intervention or no intervention. We will also include studies without a comparison group when the outcome studied is the serological response to the vaccine.

## Types of outcome measures

We will include studies that report on one or more of our primary outcomes and/or our secondary outcome of interest.

### Primary outcomes

► All-cause hospitalisation;
► All-cause mortality.

### Secondary outcomes

► Serological response to vaccine;
► Acute respiratory illness-related hospitalisation;
► Influenza illness or influenza-like-illness (ILI)-related hospitalisation (based on hospital discharge codes or clinical diagnosis);
► Pneumococcal disease-related hospitalisation;
► Hospitalisation for liver disease complications (variceal bleeding, hepatic encephalopathy, ascites, spontaneous bacterial peritonitis, jaundice and bacteraemia or sepsis);
► Acute respiratory illness-related mortality;
► Influenza illness or ILI-related mortality;
► Pneumococcal disease-related mortality;
► Liver disease-related mortality.

## Information sources

### Electronic searches

To capture all relevant studies, we plan to search the following databases:

► Cochrane Central Register of Controlled Trials;
► Medline (Ovid);
► Embase (Ovid);
► PubMed.

Each database will be searched separately and the search strategy first developed in Medline will be adapted to each database interface as appropriate. We plan to also search relevant studies from the reference lists of the eligible studies identified through the electronic searches.

## Search strategy

We will use two complementary strategies to identify relevant articles. First, we will identify articles reporting outcomes of vaccination in patients with liver disease by combining search terms for influenza and pneumococcal vaccination with search terms for chronic liver disease (search 1). This search will be filtered by study design. The provisional search terms for liver disease and pneumococcal and influenza vaccines are listed in table 1.

Recognising that liver disease patients may be included as subgroups in clinical trials of vaccination, we will also search for randomised controlled trials of influenza and pneumococcal vaccine that have recruited individuals from the general population (search 2).

To search for studies with adult participants, we will combine the geriatric and adult medicine-specific search strategies by Kastner et al[18] To maximise the sensitivity of the searches for randomised clinical trials, we will use a filter that combines terms from the Cochrane Highly Sensitive Search Strategy[19] and less specific version of the same filter by Chalmers et al[20] to identify randomised trials. Similarly, to maximise the sensitivity of searches for case–control and cohort studies, we will use a filter that combines terms from the University of Texas School of Public Health filter for observational studies,[21] Scottish Intercollegiate Guidelines Network observational study filter[22] and the BMJ Evidence Centre case-control and cohort strategy.[23] The search terms for adult and study design filters are listed in table 1. We will impose no country, setting or date restrictions on the search.

## Study records

### Data management

The search results will be uploaded into reference management software (Mendeley) to remove duplicate records of the same report. The unique records will then be uploaded into web-based, systematic review management software (DistillerSR). Both the initial abstract and title screening and the full-text review and extraction of data from the eligible studies will be performed using standardised, precreated online forms. All forms will be piloted and revised as needed by the reviewers before starting the review.

**Table 1** Medline (Ovid) provisional search terms

| Search concept | Search terms |
| --- | --- |
| Pneumococcal vaccine | 1. exp Pneumococcal Vaccines/ |
| | 2. exp Pneumococcal Infections/pc [Prevention & Control] |
| | 3. ((anti?pneum* or pneum*) adj5 (vaccin* or immuni*)).mp. |
| | 4. (PPV?23* or PPSV or PPSV?23* or PCV?7* or PCV?10* or PCV?13*).mp. |
| | 5. ((PPV or PCV) adj5 (pneum* or vaccin* or immuni*)).mp. |
| | 6. ((7?valent or hepta?valent or 10?valent or 13?valent or 23?valent) adj5 (vaccin* or immuni*)).mp. |
| | 7. 1 or 2 or 3 or 4 or 5 or 6 or 7 |
| Influenza vaccine | 1. Influenza Vaccines/ |
| | 2. Influenza, Human/pc [Prevention & Control] |
| | 3. ((anti?influenza or influenza or seasonal or anti?flu or flu) adj5 (vaccin* or immuni*)).mp. |
| | 4. ((TIV or QIV or trivalent or quadrivalent or 3?valent or 4?valent) adj5 (vaccin* or immuni*)).mp. |
| | 5. 1 or 2 or 3 or 4 or 5 |
| Liver disease | 1. exp Liver Diseases/ |
| | 2. ((liver or hepat*) adj3 disease*).mp. |
| | 3. ("chronic liver" or "chronic hepat*").mp. |
| | 4. cirrho*.mp. |
| | 5. 1 or 2 or 3 or 4 |
| Adult participants | 1. exp Adult/ |
| | 2. adult.mp. |
| | 3. (middle?aged or aged).sh. |
| | 4. age*.tw. |
| | 5. 1 or 2 or 3 or 4 |
| Randomised controlled trials | 1. randomized controlled trial.pt. |
| | 2. randomi*.ab,ti. |
| | 3. randomly.ab,ti. |
| | 4. controlled clinical trial.pt. |
| | 5. trial.ab,ti. |
| | 6. groups.ab,ti. |
| | 7. drug therapy.fs. |
| | 8. placebo.ab,ti. |
| | 9. 1 or 2 or 3 or 4 or 5 or 6 or 7 or 8 |
| | 10. Animals/ |

Continued

**Table 1** Continued

| Search concept | Search terms |
| --- | --- |
| | 11. Humans/ |
| | 12. 10 not (10 and 11) |
| | 13. 9 not 12 |
| Case–control and cohort studies | 1. Epidemiologic Studies/ |
| | 2. exp Case control studies/ |
| | 3. exp Cohort studies/ |
| | 4. Longitudinal studies/ |
| | 5. Follow up studies/ |
| | 6. Prospective studies/ |
| | 7. Retrospective studies/ |
| | 8. Control groups/ |
| | 9. Matched-Pair Analysis/ |
| | 10. (Case* adj5 control*).ti,ab,kw. |
| | 11. (Case* adj5 comparison*).ti,ab,kw. |
| | 12. Control group*.ti,ab,kw. |
| | 13. (Cohort adj (study or studies)).ti,ab. |
| | 14. Cohort anal*.ti,ab. |
| | 15. (Follow up adj (study or studies)).ti,ab. |
| | 16. (Observational adj (study or studies)).ti,ab. |
| | 17. Longitudinal.ti,ab. |
| | 18. Retrospective.ti,ab. |
| | 19. Prospective.ti,ab. |
| | 20. 1 or 2 or 3 or 4 or 5 or 6 or 7 or 8 or 9 or 10 or 11 or 12 or 13 or 14 or 15 or 16 or 17 or 18 or 19 |
| | 21. Animals/ |
| | 22. Humans/ |
| | 23. 21 not (21 and 22) |
| | 24. 20 not 23 |

### Selection process

Articles that have been identified through the broad search of RCTs of pneumococcal or influenza vaccination in the general population (search 2) will first be prescreened by title by one reviewer (SH). Articles meeting prescreening criteria will be combined with the articles identified through search 1. These articles will then all be screened by two independent reviewers (SH and CP) by abstract and title. Where the study eligibility cannot be established based on the title and abstract, the report will be passed on to the full-text review. Similarly, records subject to disagreement over eligibility will be included in the full-text review.

The full-text review will be independently completed for all eligible articles by two independent reviewers authors (SH and CP). Reasons for exclusion of ineligible studies will be recorded. Disagreements will be resolved by consulting a third review author (AO) and any uncertainties by correspondence with study investigators. Multiple reports of the same study will be collated into one and, where not possible, only the most relevant report based on our eligibility criteria will be included. The study selection process will be recorded and presented in flow diagram format according to the recommendations of PRISMA.[24]

## Data collection process

The data will be extracted and entered into standardised, precreated online data extraction forms independently and in duplicate by two review authors (SH and CP). Disagreement will be resolved by consulting a third review author (LS) and uncertainties by correspondence with study investigators.

## Data items

We will extract data on:
► Study participants: inclusion and exclusion criteria, method of recruitment/selection, study population characteristics and any imbalances at baseline (sex, age, aetiology and severity of liver disease, comorbidities, alcohol use, smoking status, prevaccination infection status, medication/treatment other than intervention);
► Interventions and comparators (vaccine type, comparison treatment, dose, route of delivery, number and timing of vaccinations/comparator treatments, number of individuals in intervention and comparison group, follow-up time in intervention and comparison groups);
► Outcomes (definition, time points measured and reported, unit of measurement, number of outcomes in the intervention and control group, unadjusted and adjusted effect measures, covariates that the effect measures were adjusted for, comparisons, missing data and reasons for missingness, statistical methods used, processes for randomisation, eg, allocation concealment);
► Study designs and methods (study type, country and setting, date of study, study duration, aim of study, withdrawals);
► Study quality and study bias (according to the needs of the assessments specified below);
► Study funding and conflicts of interest.

Effect measures will be collected in the format in which they are reported and transformed for presentation and analysis if appropriate.

## Outcomes and prioritisation

Our main outcomes of interest are hospitalisation and death. These are potential severe outcomes of influenza illness and pneumococcal disease. It may be challenging to identify and establish, especially if hospital discharge records are reviewed retrospectively, the exact cause of the hospitalisation or death of a patient with an underlying chronic condition. For this reason, our primary outcomes will include all causes. Additionally, studies may not always specify whether a patient with a diagnosis of an infectious disease or liver disease complication was actually hospitalised. Unless it is specified that the patient was hospitalised or the illness was recorded in the hospital records, we will assume the patient was not hospitalised.

Our secondary outcomes of interest are the serological response to pneumococcal and influenza vaccines and a range of cause-specific hospitalisation and mortality. Serological response to a vaccine is an indicator of the vaccine's effect on building protective immunity against the disease the vaccination targets. Since, however, it is difficult to know what level of antibody or increase in antibody concentration may provide protection in people with chronic liver disease, we will evaluate both the postvaccination antibody level and the prevaccination to postvaccination fold change in geometric mean antibody concentrations. We will include studies where blood was drawn both before vaccination and at least 2 weeks after vaccination. This effect will be evaluated both for short term (<6 months) and long term (≥6 months) after vaccination. In case antibody responses are reported at multiple short-term or long-term time points, we will consider the time point closest to the timing of vaccination as the as the short-term response and the time point closest to 6 months as the long-term response. The more detailed causes of hospitalisation and death, allow us to understand about the more specific effects of the vaccines.

## Assessment of risk of bias in individual studies

We will use the Cochrane Collaborations tool[19] for assessing the risk of bias in all studies included in the review after the full-text review. The level of risk of bias in random sequence generation, allocation concealment, blinding of participants and personnel, blinding of outcome assessment, incomplete outcome data, selective reporting and other sources will be judged as 'low', 'high' or 'unclear' according to the criteria specified in the Cochrane Handbook.[19]

In the non-randomised studies, we will additionally assess the level of risk of confounding bias due to inadequately addressed differences between groups (ie, is the effect estimate likely to be biased due to unaccounted confounding). We will consider age, sex, severity and aetiology of liver disease to be the most important potential confounders. We will judge the risk of confounding bias to be low if the study addressed the presence of these confounders by restricting participant selection by confounders, demonstrating balance between groups, matching on the confounders or adjusting for the confounders in statistical analyses of the effect size. The risk of confounding bias will be judged low if the confounders were adjusted for, high if the presence of

the confounders was not addressed and unclear if there was insufficient information to judge the level of risk.

Two review authors (SH and CP) will independently assess the studies for each of the risk areas by entering a quote from the study to describe the procedures, their judgement together with a justification of the judgement into precreated online forms in DistillerSR. Disagreements will be resolved by consulting a third review author (AH). The risk of bias assessments will be presented in a figure that shows the level of risk in the different risk areas within each individual study and in a graph that describes the proportion of studies within each risk level per risk area.

### Assessment of bias in conducting the systematic review

We will conduct the systematic review following this prespecified protocol and report any differences between the methods of the complete review and this protocol in the review.

### Data synthesis

#### Criteria for quantitative data synthesis

We plan to carry out a formal meta-analysis only where more than a single study per outcome is identified and the study designs, protocols and measures of treatment effect are considered similar enough to produce a meaningful pooled effect.

#### Measures of treatment effect

For dichotomous data, the treatment effect will be estimated and presented as a risk ratio (RR) with 95% CI. For time-to-event data, we will present the results as a log HR with its SE. For studies reporting on serological response using a cohort design without a comparison group, we will present the effect of vaccination as the postvaccination antibody level and the prevaccination to postvaccination fold change in geometric mean antibody concentration.

#### Unit of analysis issues

The outcomes will be analysed at the level of study participants from each individual study.

#### Dealing with missing data

We will contact investigators to obtain numerical outcome data that have not been fully reported (for instance where when a study is identified as an abstract only or outcomes are reported in figures only). Where possible, we will calculate missing SD from other reported statistics such as CIs or SEs. The impact of including studies with high levels of missing outcome data on the treatment effect will be explored in the sensitivity analysis.

#### Assessment of heterogeneity

To assess heterogeneity between studies, we plan to present a forest plot for each of the review outcomes. We will then calculate the formal heterogeneity variance statistics $\tau^2$, $I^2$ and the Q-statistic. We will regard heterogeneity as substantial if $\tau^2$ is greater than 0, $I^2$ is more than 30% and the P value for Q-statistic is less than 0.10. We

plan to further explore the potential causes of substantial heterogeneity in the subgroup analyses or meta-regression (specified below).

#### Quantitative data synthesis

Statistical analyses will be performed using Stata 14 or R Studio. To account for the presence of heterogeneity, we will use random-effects meta-analysis to summarise the average effects of vaccination on the defined outcomes across studies. The results will be presented in forest plots with the average treatment effect (RR) with 95% CI, and the estimates of $\tau^2$ and $I^2$. We will use the pooled average treatment effect to calculate the effectiveness of the vaccines (100*(1-RR)) in preventing the primary outcomes. The immunogenetic effect of the vaccines will be summarised as the mean post-vaccination antibody level with 95% CI and the mean pre- to post-vaccination fold change in geometric mean antibody concentration with 95% CI. Observational studies and randomised controlled trial studies will be considered in separate analyses.

#### Subgroup analysis and investigation of heterogeneity

In case we identify an adequate number of studies (studies per explanatory variable ≥10), we plan to investigate the potential causes of heterogeneity between studies through random effects metaregression analyses. We will consider the following categories of explanatory variables: severity of liver disease, aetiology of liver disease and the reason for hospital admission/mortality (primary outcomes). Inclusion/exclusion of the explanatory variables in the heterogeneity investigations will depend on the characteristics and design of the identified studies. If we do not identify enough studies to perform metaregressions but there are a minimum five studies per analysis, we plan to carry out subgroup analyses to investigate whether these explanatory variables can explain heterogeneity between the studies.

#### Sensitivity analysis

In case the identified studies differ in terms of risk of bias, we plan to investigate the impact of excluding studies with high/unclear risk of bias on effect estimates in sensitivity analyses.

#### Qualitative data synthesis

We will provide a narrative summary of the study results for all outcomes, categorised by study design and vaccine type (influenza vaccine and pneumococcal vaccine). For the primary outcomes, we will report the cause of hospitalisation and death studied. Characteristics (participants, interventions, comparators, outcomes, study design and methods and notes on funding and conflicts of interests) of all studies included in the review will also be presented in separate tables. The results for outcomes where meta-analysis will not be carried out due to insufficient homogeneity between studies will be presented in forest plots without the pooled effect estimate.

## Metabias(es)

### Assessment of reporting biases across studies

We plan to investigate reporting bias using funnel plots. If there are enough studies in the analysis (minimum 10), we will also carry out the Egger's test to assess whether there is a linear association between the study's result and its SE.

We plan to assess selective outcome reporting bias by comparing what the study set to measure and analyse in the Methods section of the study report (for studies published after 2006, we will also investigate the details trial protocol if it can be identified through the WHO International Clinical Trials Registry Platform,[25] launched in 2007) with the results that were reported. Using the Outcome Reporting Bias in Trials classification system,[26] we will evaluate whether the risk of selective outcome reporting bias is present and whether the risk is low or high.

### Confidence in cumulative evidence

We will use the Grading of Recommendations Assessment, Development and Evaluation Working Group system[27] to assess and report the overall quality of the body of evidence for each outcome studied. The within-study risk of bias (methodological quality), directness of evidence, heterogeneity, the precision of effect estimates and risk of publication bias will be independently assessed by two review authors (SH and CP). The quality of evidence will be judged and reported as 'high', 'moderate', 'low' or 'very low' following the Cochrane Handbook for Systematic Reviews of Interventions guidelines.[19]

**Contributors** The study was conceived by SH, LS, AO and AH. SH developed the eligibility criteria, search strategy, risk of bias assessment strategy and data extraction plan with guidance from LS, AO and AH. SH wrote the manuscript, to which all authors contributed.

**Funding** This work was supported by the UK Biotechnology and Biological Sciences Research Council grant number BBSRC BB/M009513/1 to SH.

**Competing interests** None declared.

**Patient consent** Not required.

**Ethics approval** Formal ethical approval is not required for this study, as no primary data will be collected.

**Provenance and peer review** Not commissioned; externally peer reviewed.

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
