## [Reviewer comments · BMJ Open]

ARTICLE DETAILS

TITLE (PROVISIONAL)	EFFECTIVENESS OF PNEUMOCOCCAL AND INFLUENZA VACCINES TO PREVENT SERIOUS HEALTH COMPLICATIONS IN ADULTS WITH CHRONIC LIVER DISEASE: A PROTOCOL FOR A SYSTEMATIC REVIEW
AUTHORS	Härmälä, Suvı; Parisinos, Constantinos; Shallcross, Laura; O'Brien, Alastair; Hayward, Andrew

VERSION 1 – REVIEW

REVIEWER	Veronica Loy Loyola University Medical Center- United States
REVIEW RETURNED	05-Jul-2017

GENERAL COMMENTS	I think this plan is excellent, however the review would be better received if you waiting until after you completed the review to publish rather than publishing the protocol alone.
---

REVIEWER	Marc-Alain Widdowson Centers for Disease Control and Prevention CDC Kenya Nairobi, Kenya
REVIEW RETURNED	05-Nov-2017

GENERAL COMMENTS	This manuscript describes a protocol for systematic review of the effectiveness of pneumococcal and influenza vaccines in persons with chronic hepatitis. Overall the paper and the methodology is generally clearly explained, and the question is relevant - it is important to understand the effectiveness of vaccines in different vulnerable populations such as this who are often older and that may have higher burden of bacterial and viral respiratory infections. I do have some concerns and questions however on the methods. 1) I suspect that there will be very little literature that will have meaningful data on the effectiveness of these vaccines in persons with chronic hepatitis specifically and then to assess if is any different to what one would expect in persons without hepatitis – which is the implicit aim of the exercise. Some rationale as to why these vaccines might not work well would be welcome in the introduction. 2) I am concerned about the primary outcomes to be assessed – all cause hospitalizations and deaths. These outcomes have been
--

	notoriously difficult to assess for influenza vaccines at least, and in large part because of bias, but also because the power to detect an effect – studies need to be very large and unlikely to be the case in a RCT. I also am curious as to why the authors did not include VE studies with laboratory- confirmed mild outcomes as either primary or secondary outcomes. There are many studies of VE for lab-confirmed outcomes (the gold standard for influenza VE studies) and especially for Influenza-like illness. If the primary hypothesis is to see if these vaccines work well in the population of interest, looking at the effect on mild illness would help understand this—in a similar way that looking at immunogenicity (as included in the protocol) might- and would increase the number of informative studies. Actually on page 7 line 47, the authors, say that they will look at the effect of pneumococcal and influenza vaccines on “preventing infection” 3) I would suggest that (as appropriate and possible) the authors consider additional confounders for observational studies, such as other comorbidities, the healthy vaccinee bias etc. 4) Would be useful to specify if adjuvanted, high-dose and intradermal vaccines were considered when building the search strategy. It seems that recombinant influenza vaccines (FluBlok) were not included which might be an omission. Though not all these are licensed in Europe, it may be useful to include any pertinent study to expand the sources of data. 5) Did the authors consider Chinese language papers? Hepatitis B virus infection has been an ongoing public health problem and China manufactures influenza vaccines.
--	---

VERSION 1 – AUTHOR RESPONSE

Dear Editor, Dear Reviewers,

Thank you for taking the time to review our study protocol and for your comments.

We are planning to publish the protocol first so that what was originally planned is clearly recorded. We will also publish the completed review as suggested by Dr Loy and what we did in the review can then easily be compared with what we planned to do as the protocol was published.

As suggested by Dr Widdowson (comment n.1), we have now added a sentence about the possibility of vaccines not working in liver disease population to the introduction section (p.5, lines 39-40)

Regarding the outcomes of this review (Dr Widdowson's comment n.2), we selected all-cause outcomes to be the primary outcomes as it can be challenging to identify the cause of hospitalisation or death in patients with a chronic disease especially at an advanced stage. We suspected that many of the studies we find may be record-based observational studies as not vaccinating people who may be at high risk of severe outcomes without vaccination can be ethics-wise questionable. While this review will be focused on assessing severe health outcomes with serological outcomes as supportive outcomes, we will definitely consider looking at the milder outcomes in the future and will mention this in the discussion of the actual review. We have corrected the sentence on page 7, line 47, of the protocol to reflect the actual outcomes of this review.

Considering additional confounders to the reviewer was suggested (Dr Widdowson's comment n.3). We have added previous vaccinations near the start of the study as an additional confounder in the review. Previous vaccinations near the study start may boost/dampen the response of the vaccine studied. We have not added this to the protocol as this is something added later on. We will specify the differences between our protocol and the review in our review.

Adjuvanted, high-dose and intradermal vaccines were included, and as suggested by Dr Widdowson (comment n.4), we have revised the protocol to be clear on this (p.11, lines 30-31). We decided, however, to exclude experimental vaccines and recombinant vaccines. This has now also been specified in the protocol (p.8, lines 30-31).

Our review unfortunately has time and budget constraints and so we were not able to consider translation services and looking into Chinese language papers (as suggested in Dr Widdowson's comment n.5).

In addition to the above changes to the manuscript, we have corrected a few errors in the text. These are: 1) in the abstract, lines 16-21, and p. 12, lines 37-47, we have corrected the list of data sources as with original list we in practice included Cochrane Trial Register multiple times (the records specific to diseases are also included in the full register), 2) p.8, line 30, we have removed the names of the influenza B lineages as this detail is not informative, 3) p.10, lines 45-47, we have corrected the sentence to reflect the actual outcomes of the review (hospitalisation and death), 4) p.21, lines 34-35, we have removed the REML method as this is not correct and finally, 5) on page 24, line 24, we have specified that the funding associated with this work is to SH (the corresponding author).

PRISMA-P checklist with page numbers is submitted with the revised manuscript, PROSPERO protocol registration is included both in the abstract and on page 21 (immediately before the References section) of the main text, and we have formatted the manuscript as per the guidelines for protocol articles.

Sincerely yours,

Suvi Härmälä

VERSION 2 – REVIEW

REVIEWER	Marc-Alain Widdowson US Centers for Disease Control and Prevention Kenya
REVIEW RETURNED	15-Jan-2018

GENERAL COMMENTS	Thank you to the authors for addressing some of my comments. I still though have some remaining suggestions. 1) The issue of outcomes, I still find a little perplexing. None of the outcomes (primary or secondary) include laboratory confirmed influenza or pneumococcal disease (either via radiograph or blood culture). It is true as the authors state that RCTs with placebo are few are far between (though there are couple) because of ethical issues, but many case control studies of VE for flu will start with a influenza confirmed case of hospitalized respiratory disease (SARI). It is puzzling to me why to exclude these specific outcomes from the
--

	outset. However, I accept that the authors have given due consideration to the issue and it is not incorrect perse, but may limit the number of relevant articles. 2) Related to above, if observational studies of discharge data are included, I do encourage checking for confounder for more than just liver disease as a confounder. Many of these patients will have other comorbidities that will affect their risk of serious outcomes and of vaccination. 3) The secondary outcomes mention “influenza illness” but not clear what that is. Do the authors mean hospital discharge codes for influenza here? Please clarify. 4) For the serologic outcomes can the authors specify criteria for inclusion such as blood taken before vaccination and at least X weeks after. This would exclude any studies that just have single blood draws after vaccination.
--	---

VERSION 2 – AUTHOR RESPONSE

Dear Editor, Dear Reviewers,

Thank you for reviewing our revised manuscript and for your comments and suggestions on how it could be further improved. Please find our point-by-point response below (each reviewer comment is followed by our response).

Reviewer comment:

1) The issue of outcomes, I still find a little perplexing. None of the outcomes (primary or secondary) include laboratory confirmed influenza or pneumococcal disease (either via radiograph or blood culture). It is true as the authors state that RCTs with placebo are few are far between (though there are couple) because of ethical issues, but many case control studies of VE for flu will start with a influenza confirmed case of hospitalized respiratory disease (SARI). It is puzzling to me why to exclude these specific outcomes from the outset. However, I accept that the authors have given due consideration to the issue and it is not incorrect perse, but may limit the number of relevant articles.

Our response: The main aim of our review is to understand the effects of the vaccines to prevent hospitalisation and death. The motivation behind focusing the outcomes to all-cause and clinical diagnosis/hospital discharge-code based outcomes is that many cases (especially flu) are not confirmed by laboratory tests and thus, lab-confirmed outcomes will greatly underestimate the burden of disease and the impact of vaccination on this burden. As Dr Widdowson points out, not including the lab-confirmed outcomes may, however, limit the number of studies our review can identify. This will be acknowledged in the discussion of our review.

Reviewer comment:

2) Related to above, if observational studies of discharge data are included, I do encourage checking for confounder for more than just liver disease as a confounder. Many of these patients will have other comorbidities that will affect their risk of serious outcomes and of vaccination.

Our response: As suggested by Dr Widdowson, we have now added the presence of co-morbidities as important confounders. We also consider the effect of the measurement period to account for the presence of confounding resulting from unmeasured differences between vaccinated and unvaccinated individuals. A significant effect seen not just within but also outside the influenza

circulation in the same study would indicate that the effect may not be attributable to the vaccine alone. The same may also be the case in studies of pneumococcal vaccine as seasonality with influenza exists especially in the elderly population. Addition of these confounders will be specified in the review itself (differences between the protocol and review section) to provide the reader with a clear track of how the protocol evolved.

Reviewer comment:

3) The secondary outcomes mention “influenza illness” but not clear what that is. Do the authors mean hospital discharge codes for influenza here? Please clarify.

Our response: All clinical outcomes in our review are hospitalisation or mortality outcomes. This specific secondary outcome is hospitalisation related to influenza (hospital discharge codes) or influenza-like illness (based on clinical diagnosis). We anticipate that the former may be used in observational and the latter in experimental studies. As suggested by Dr Widdowson, this has now been specified in the protocol (p. 9 line 6 of the marked copy).

Reviewer comment:

4) For the serologic outcomes can the authors specify criteria for inclusion such as blood taken before vaccination and at least X weeks after. This would exclude any studies that just have single blood draws after vaccination.

Our response: We also thank Dr Widdowson for the suggestion to clarify the criteria for inclusion in terms of the serological responses. We will include studies where blood was drawn before vaccination and at least 2 weeks after vaccination. This has now been specified in the protocol (p. 15 lines 49-50 of the marked copy).

In addition to the above changes to the manuscript, we have removed a sentence saying that we will assume that a patient was taken to hospital in case they had a liver disease complication or invasive pneumococcal disease (p. 15 lines 19-25 of the marked copy). This is incorrect and we will not assume a patient being admitted to hospital unless it is clear that the hospitalisation happened.

Sincerely yours,

Suvi Härmälä

VERSION 3 – REVIEW

REVIEWER	Marc-Alain Widdowson CDC-Kenya Centers for Disease Control and Prevention Kenya
REVIEW RETURNED	07-Feb-2018

GENERAL COMMENTS	No further comments.
----------------------